# Cross-species regulatory sequence activity prediction

**David R. Kelley**  *

Calico Life Sciences, South San Francisco, California, United States of America

* drk@calicolabs.com

## Abstract

Machine learning algorithms trained to predict the regulatory activity of nucleic acid sequences have revealed principles of gene regulation and guided genetic variation analysis. While the human genome has been extensively annotated and studied, model organisms have been less explored. Model organism genomes offer both additional training sequences and unique annotations describing tissue and cell states unavailable in humans. Here, we develop a strategy to train deep convolutional neural networks simultaneously on multiple genomes and apply it to learn sequence predictors for large compendia of human and mouse data. Training on both genomes improves gene expression prediction accuracy on held out and variant sequences. We further demonstrate a novel and powerful approach to apply mouse regulatory models to analyze human genetic variants associated with molecular phenotypes and disease. Together these techniques unleash thousands of non-human epigenetic and transcriptional profiles toward more effective investigation of how gene regulation affects human disease.

## Author summary

Human population genetic studies have highlighted thousands of genomic sites that correlate with traits and diseases that do not modify gene sequences directly, but instead modify where and when those genes are expressed. To better understand how these sites influence traits and diseases, and consider their relevance for drug development, we need better models for how DNA sequences determine gene expression. Recently, machine learning algorithms based on deep artificial neural networks have proven to be promising tools toward this end. In this work, we improve upon the state of the art model accuracy by combining training data from both humans and mice. Using these models, we can predict the effect of a genetic variant on gene expression in any tissue or cell type with available data. We further demonstrate that predictions for human variants derived from mouse training datasets are highly informative and offer unique insight into the genetic basis of gene expression and disease.

**Data Availability Statement:** All data used in this study are publicly available via ENCODE, GEO, and FANTOM. S1 Table describe all data and sources.

**Funding:** The author received no specific funding for this work.

**Competing interests:** DRK is employed by Calico LLC.

## Introduction

Predicting the behavior of any nucleic acid sequence in any nuclear environment is a primary objective of gene regulation research. In recent years, machine learning approaches to directly tackle this problem have achieved significant accuracy gains predicting transcription factor (TF) binding, chromatin features, and gene expression from input DNA sequence [1–7]. These models have been applied to study genetic variation in populations and generate mechanistic hypotheses for how noncoding variants associated with human disease exert their influence [3–8]. Estimates for how mutations influence regulatory activity have also offered novel views into regulatory evolution and the robustness of genes to such mutations [7].

Deep convolutional neural networks have achieved state of the art performance for many regulatory sequence activity prediction tasks in humans and other species [2, 4–7]. The complexity of mammalian gene regulation and these models' impressive but imperfect predictions suggest room for improvement remains. In particular, distal regulation by enhancers is incompletely captured by existing models, which do not attempt to consider sequence beyond ~20 kb of transcription start sites (TSSs) [5, 7, 9]. Obtaining more training data is a reliable strategy to improve model accuracy. The research field continues to generate new functional genomics profiles, but these merely deliver additional labels for the existing sequence data; fitting more expressive and accurate models would benefit more from entirely new training sequences. Acquiring functional profiles for more humans will provide limited new training data because individual human genomes differ only slightly from each other. In reporter assays, large quantities of synthetic sequences can be profiled, but they are limited to short sequences and cell lines that cannot represent the full complexity of human tissues [10–14].

Non-human species offer a potential source of this desired additional training data. Regulatory sequence evolves rapidly, but TF binding preferences are highly conserved due to the drastic effect that modifying affinity for many thousands of binding sites would confer on the organism [15–17]. Prior work studying cross-species machine learning of TF binding and chromatin marks indicates potential utility [18, 19]. Thus, we hypothesized that regulatory grammars across related species have enough in common that jointly training large models on vast data from multiple species will improve regulatory sequence activity models derived by machine learning. To explore this data source, we chose the mouse as a distant mammal with substantial functional genomics data available [20].

In addition to serving as a source of more genomic sequences, mouse experiments can explore biological states that are difficult or unethical to acquire in humans, e.g. profiling mouse development, disease, and genome modifications. If context-specific regulatory grammars are sufficiently conserved across species, then models trained to predict these mouse data may be able to impute human genome profiles to study human regulatory sequences and genetic variation. Although models trained on mouse will not match the performance of analogous human models, they may serve as useful approximations and produce novel variant annotations when the human data are unavailable.

In this work, we trained a deep multi-task convolutional neural network to jointly learn the complex regulatory grammars that determine TF binding, chromatin marks, and transcription using the ENCODE and FANTOM compendia of thousands of functional genomics profiles from hundreds of human and mouse cell types. We benchmarked single versus joint genome training and found that jointly training on human and mouse data leads to more accurate models for both species, particularly for predicting CAGE RNA abundance. We demonstrated that mouse regulatory grammars can be transferred across species to human where they continue to make accurate tissue-specific predictions. Applying this procedure to predict human

genetic variant effects revealed significant correspondence with eQTL statistics and proved insightful for studying human disease.

## Results

### Multi-genome training improves gene expression prediction accuracy

We applied the Basenji software and framework to predict functional genomics signal tracks from only DNA sequence [5]. The neural network takes as input a 131,072(= $2^{17}$) bp sequence, transforms its representation with iterated convolution layers, and makes predictions in 128 bp windows across the sequence for the normalized signal derived from many datasets (Fig 1, Methods). We applied an architecture that uses residual connections to alleviate the strain of vanishing gradients in deep network optimization and improve generalization accuracy (S1 Fig) [21]. Training on multiple genomes required several further developments (Methods). Most importantly, we modified the train/valid/test split of the genomic sequences to ensure that homologous regions from different genomes did not cross splits (Methods); without this step, we might overestimate generalization accuracy.

We assembled training data consisting of 6,956 human and mouse quantitative sequencing assay signal tracks from the ENCODE and FANTOM consortiums (Methods). These data describe regulatory activity across tissues and isolated cell types using several techniques—DNase and ATAC-seq to measure DNA accessibility, which typically mark TF-bound sites, and ChIP-seq to map TF binding sites and histone modification presence [22, 23]. The FANTOM data consists of RNA abundance profiling with CAGE, where the 5' end of the transcript is sequenced [24]. These 5' RNA profiles are independent of splicing and allow us to provide input DNA sequence without gene annotations, which would not be the case for RNA-seq [5]. In addition, we added several mouse datasets describing cell states that are unavailable for humans: (1) a single cell ATAC-seq atlas from 13 tissues clustered to 85 distinct profiles [25] and (2) several TF and chromatin profiles obtained over 24 hour time courses in the liver to study circadian rhythms (S1 Table).

To measure the influence of multi-genome training on generalization accuracy, we trained three separate models on these data: one jointly fit to both human and mouse, one to human data alone, and one to mouse data alone. For each scenario, we fit the same model architecture and hyperparameters. We trained the models to minimize log Poisson loss and assessed accuracy by computing Pearson correlation between the predictions and observed signal for each dataset.

The joint training procedure improved test set accuracy for 94% of human CAGE and 98% of mouse CAGE datasets (binomial test p-values $1 \times 10^{-16}$ and $1 \times 10^{-16}$), increasing the

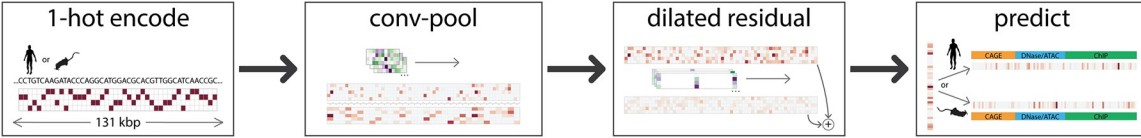

**Fig 1. Predicting regulatory sequence activity for human and mouse genomes.** We predict the regulatory activity of DNA sequences for multiple genomes in several stages (Methods). The model takes in 131,072 bp DNA sequences, encoded as a binary matrix of four rows representing the four nucleotides. We transform this representation with seven iterated blocks of convolution and max pooling adjacent positions to summarize the sequence information in 128 bp windows. Green and purple heatmaps represent convolution filter weights; red and white heatmaps represent pooled sequence vectors. To share information across the long sequence, we apply eleven dilated residual blocks, consisting of a dilated convolution with exponentially increasing dilation rate followed by addition back into the input representation. Finally, we apply a linear transform to predict thousands of regulatory activity signal tracks for either human or mouse. All parameters are shared between species except for the final layer.

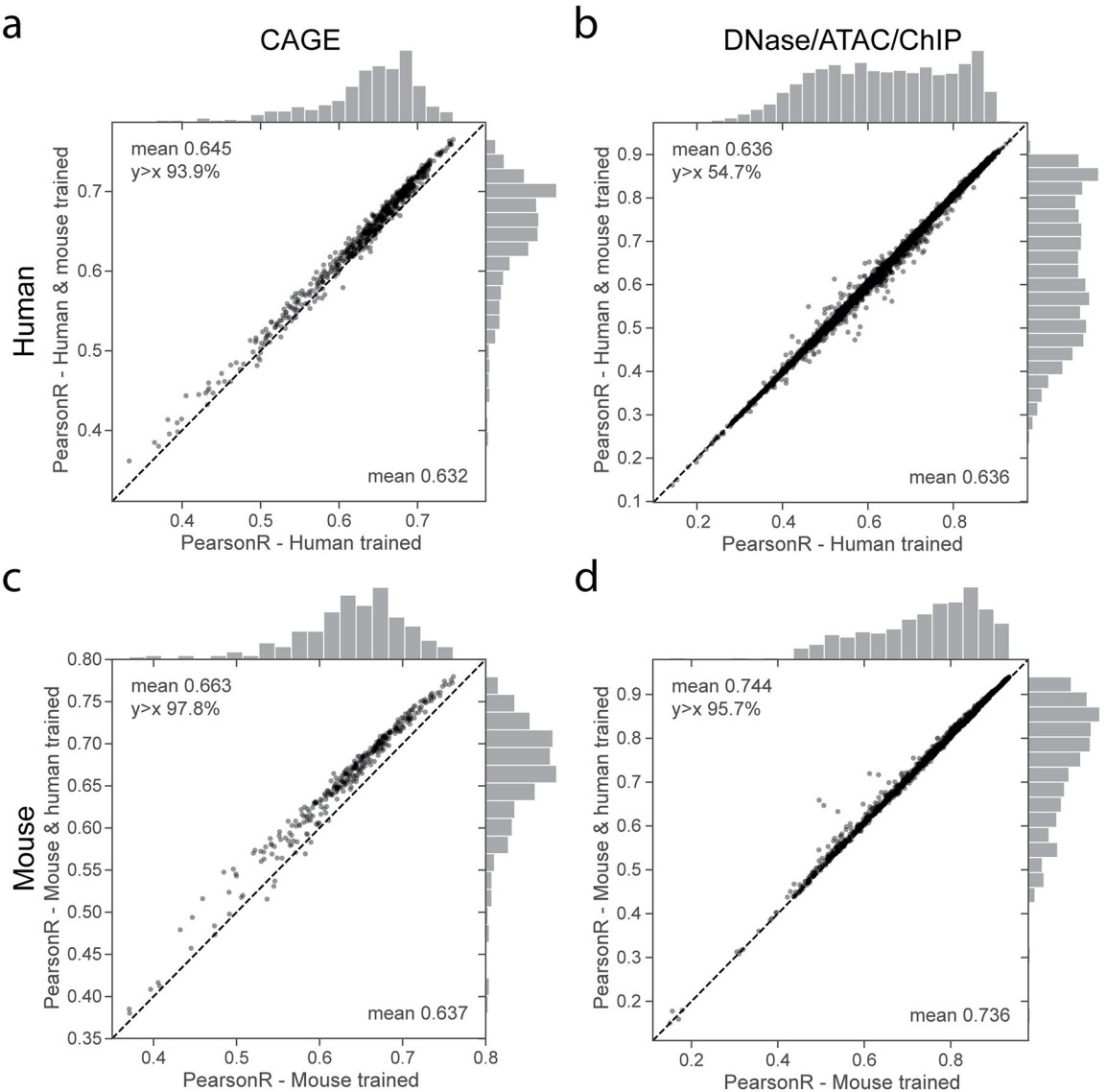

**Fig 2. Training on human and mouse data improves generalization accuracy.** We trained three separate models with the same architecture on human data alone, mouse data alone, and both human and mouse data jointly. For each model, we computed the Pearson correlation of test set predictions and observed experimental data for thousands of datasets from various experiment types. Points in the scatter plots represent individual datasets, with single genome training accuracy on the x-axis and joint training accuracy on the y-axis. For CAGE, training on multiple genomes increases test set accuracy on nearly all datasets for both (a) human and (c) mouse. (b,d) For DNase/ATAC/ChIP-seq, test set accuracy improves by a smaller average margin. See S3 Fig for additional splits by assay and ChIP immunoprecipitation target.

average correlation by .013 and .026 for human and mouse respectively (Fig 2a and 2c). For DNase, ATAC, and ChIP, joint training improved predictions by a lesser margin relative to single genome training; average test set correlation increased for 55% of human and 96% of mouse datasets (binomial test p-values $3 \times 10^{-11}$ and $1 \times 10^{-16}$) (Fig 2b and 2d).

Comprehensive breakdown across the various classes of functional genomics experiments revealed heterogeneous benefits across the ChIP-seq immunoprecipitation targets (S3 Fig). Predictions for the promoter mark H3K4me3 generally improved, but accuracy for the hetero-chromatin mark H3K9me3 decreased slightly on average across cell types (S3 Fig), perhaps

due to human and mouse genomes containing different repetitive elements that are silenced by the modification. We made use of annotations assigned to the human DNase collection by Meuleman et al. to split DNase accuracy comparisons by fifteen distinct organ systems [26]. Although several systems achieved significantly lesser or greater accuracy margins, deviations were small, and no clarifying patterns emerged (S4 Fig). The datasets where single genome accuracy exceeded joint tended to be consistent across a second independent batch of training runs (S5 Fig). ChIP-seq experiments in the cancer cell lines K562 and MCF-7 were significantly enriched in this set, suggesting that modeling these datasets may slightly suffer due to absence of an analogue across species or somatic mutation divergence from the reference genome.

Joint genome training delivers both more sequences and more annotations. To clarify which factor is more important for improving accuracy, we conducted additional training experiments in which we only modulated the number of human genome annotations. In this regime of ample data, holding out even half of the human datasets to reduce the number trained on did not reduce predictive performance (S6 Fig). This implies that the joint model's increased accuracy depends on also having the additional training sequences contributed by the second genome.

CAGE has several properties that may explain the observed benefit of having more training data from multiple genomes. CAGE signal has a larger dynamic range that spans orders of magnitude and more sophisticated transcriptional regulatory mechanisms that often involve distant sequences. Measured by gradient-based saliency analysis, the jointly trained model makes greater use of long range activating elements ($> 10$ kb) to predict CAGE signal at TSSs (S2 Fig). Altogether, these results demonstrate that regulatory grammars are sufficiently similar across the 90 million years of independent evolution separating human and mouse so that their annotated genomic sequences provide informative multi-task training data for building predictive models for both species.

## Regulatory sequence activity models transfer across species

Regulatory activity conservation across related species has been observed in genome-wide functional profiles of TF binding and histone modifications [15–17]. In matched tissue samples, similar TFs are typically present and those TFs have highly conserved motif preferences [17, 27]. These findings suggest that a regulatory sequence activity model trained to predict for one species will also make usefully accurate predictions for matched samples from the other. This was recently demonstrated for enhancer annotations and histone modifications across a variety of mammals [18]. To quantify this phenomenon for our models and data, we selected several diverse and representative tissues and cell types for which we could unambiguously match across species—cerebellum, liver, and CD4+ T cells. We extracted CAGE gene expression measurements from the TSSs for all human genes outside the training set and computed predictions for human and mouse versions of these tissues and cell types (Fig 3a). For this exercise, and those to follow, we used the jointly trained multi-task model and sliced out predictions of interest, but results were consistent for the model trained only on mouse data (S7 Fig).

Across human gene TSSs, we observed cross-species prediction accuracy of 0.73 Pearson correlation for mouse predictions to human observed signal averaged across these samples. This approaches the 0.75 correlation for human predictions to human observed signal, but does suggest that some genome-specific activity exists (S8 Fig). To assess whether the model further captures and transfers tissue specificity, we normalized each TSS's data or predictions by its mean across all CAGE datasets. Mean normalization removes correlation driven by

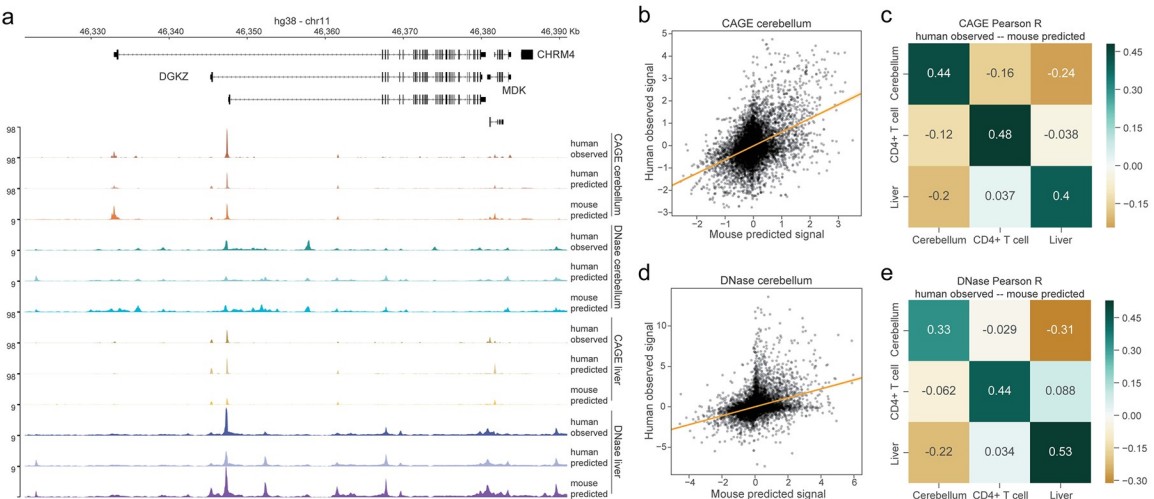

**Fig 3. Regulatory grammars are largely conserved across species.** (a) Tissue-specific regulatory grammars can be learned and transferred across species, exemplified here by CAGE and DNase data and predictions for cerebellum and liver. The "human predicted" tracks describe predictions for the human datasets displayed as "human observed"; "mouse predicted" tracks describe predictions for the matched mouse dataset. We scaled coverage tracks by their genome-wide means separately within all CAGE and all DNase/ATAC data. (b,d) Mouse predictions for cerebellum CAGE and DNase correlate strongly with human data. For CAGE, points represent the top 50% most variable TSSs. Data or predictions were quantile normalized to align sample distributions, log transformed, and mean-normalized across samples. For DNase, points represent the top 10% most variable genomic sites (less than CAGE because we consider the whole genome rather than TSSs). Data or predictions were similarly quantile normalized to align sample distributions and mean-normalized across samples. The statistical trends were robust to top variable threshold choice. Scatter plot lines represent ordinary least squares regressions. (c,e) These correlations are specific to the matched tissues and not shared by others.

accurate prediction of global cross-tissue activity. On this more challenging task, normalized mouse predictions achieved mean 0.40 Pearson correlation with normalized human data for the matched samples (Fig 3b and 3c). In contrast, normalized predictions compared to data from distinct tissues/cell types resulted in negative or near zero correlations (Fig 3c). Thus, the model has learned tissue and cell type specificity beyond a baseline level and is able to transfer that knowledge across species.

We repeated these analyses with DNase accessibility profiles for the same tissues and cell types to assess how general this transferability is for different data. Because most sites lack activity, we selected the top 10% most variable. We observed the same statistical trends for accessibility—high correlation between mouse predictions and human data for matched samples (mean 0.84) and specificity in scaled comparisons (Fig 3d and 3e).

Less correlation for the mouse predictions relative to human predictions is expected because the human predictions are derived from training on the exact experimental sample (but different sequences), which may have its own unique attributes and biases. To gain further insight into this phenomenon, we plotted the human and mouse residuals against local GC% in a 1,000 bp window around the sites (S9 Fig). Indeed, we observed correlations between GC% and the residuals that were greater for the mouse versus human predictions. Thus, GC% calibration, or more sophisticated forms of domain adaptation, may be an interesting avenue to explore to further improve cross-species transfer [28].

## Mouse-trained models elucidate human genetic variant effects

A driving goal of regulatory sequence modeling is to predict the effect of human genetic variants on gene expression and downstream phenotypes. For any biallelic variant, we can predict signal across the surrounding genomic sequence for each allele and derive a summary score

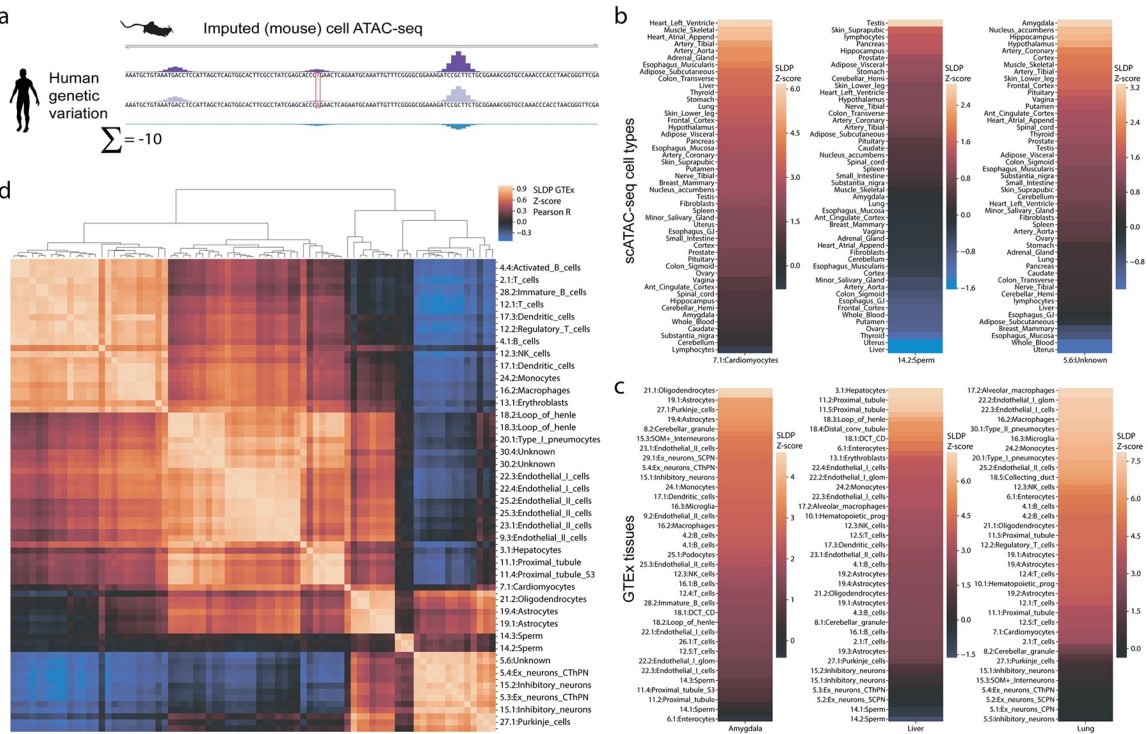

**Fig 4. Mouse cell type accessibility predictions show a strong and specific statistical relationship with human eQTLs.** (a) We predicted the effect of human genetic variants on imputed regulatory signal trained on mouse single cell ATAC-seq (scATAC) cluster profiles. We scored variants by subtracting the signal from the minor allele from that of the major and summing across the sequence. (b) We used signed linkage disequilibrium profile (SLDP) regression to compare the cell type-specific variant effect predictions to tissue-specific eQTL summary statistics from GTEx. Cell type profiles correspond best with the expected tissues. (c) GTEx tissues correspond best with the expected cell types. (d) Clustering scATAC cell types by their Z-scores across GTEx tissues reveals the expected structure.

for the variant effect (Fig 4a). Here, we sum the signal across the sequence and take the difference between alleles. We can compute this score for every dataset using two forward passes of the convolutional neural network.

Models trained on mouse data allow one to predict the difference between how two human alleles would behave if they were present in the regulatory environment of mouse cells. Given the evidence that analogous human and mouse cells largely share regulatory grammars, we hypothesized that models trained on mouse data would be insightful towards understanding human regulatory variants' function. To test this hypothesis, we studied the Gene-Tissue Expression (GTEx) release v7a data of genotypes and gene expression profiles for hundreds of humans across dozens of tissues [29]. In previous work, we showed that variant scores derived from Basenji predictions corresponded significantly with GTEx summary statistics [5]. Here, we conducted a similar analysis using signed linkage disequilibrium profile (SLDP) regression to measure the statistical concordance between signed variant effect predictions and GTEx summary statistics (Methods) [8]. SLDP distributes a signed annotation (i.e. our scores) according to a given population's LD structure and compares to a set of summary statistics. Using a permutation scheme, the method produces a signed Z-score that specifies the direction and magnitude of the relationship and a p-value describing its significance.

We focused on a dataset unique to the mouse—a single cell ATAC-seq atlas from 13 adult mouse tissues that reported 85 distinct cell types after clustering analysis [25]. Merging aligned reads for each cell type cluster produced pseudo-bulk coverage profiles, which we trained to

predict. We sliced predictions for these datasets from the model trained jointly on all human and mouse data. We first asked whether coverage tracks derived from clustering single cell assays are amenable to Basenji modeling. Predictions for held out sequences achieved Pearson correlation ranging from 0.43-0.84 in 128 bp windows for these 85 profiles, which is in line with predictions for bulk DNase/ATAC-seq.

Human variant predictions for these models generally exhibited a strong positive relationship on GTEx summary statistics, in line with prior observations that accessibility correlates with gene expression. Furthermore, cell type predictions aligned well with anatomical expectations. For example, variant predictions for cardiomyocytes have the strongest relationship with GTEx measurements in the heart and skeletal muscle (Fig 4b). From the opposite direction, GTEx measurements for the liver have the strongest relationship with variant predictions for hepatocytes (Fig 4c). These results further support the claim that human and mouse cells share relevant regulatory factors and that our procedure can project these factors across species from mouse experiments to human variants. For each pair of mouse ATAC cell types, we computed the correlation between their SLDP Z-scores across GTEx tissues (Fig 4d). The correlations revealed expected structure, with clusters representing the blood, endothelial cells, neurons, among others.

Next, we asked whether these or any other mouse datasets were informative above and beyond available human datasets. Theoretically, we can add scores for every human dataset to the SLDP background model, forcing the statistical test for mouse dataset scores to consider only the residual variance in GTEx summary statistics (Methods). We implemented a far more computationally efficient close approximation, in which we added 64 principal components of the variant by score matrix for human datasets, which explained 99.9% of variance for CAGE and 99.3% for DNase/ATAC.

Even considering the human data, many mouse datasets still emerged as delivering orthogonal value by SLDP (S10 Fig). Among CAGE data, developmental heart profiles from neonate and embryo stages had significant positive relationships with GTEx tibial artery (10 datasets with FDR $q < 0.05$) and left ventricle (41 datasets with FDR $q < 0.05$). These suggest that human adult heart gene expression depends on genetic variation that acts more prominently during early development, which is recovered more effectively by current data from mice. Among DNase/ATAC, variant predictions for the single cell hepatocytes and a 24 hour time course to profile circadian rhythms of genome accessibility in the liver [30] showed a significant positive relationship with the GTEx liver statistics (7 datasets with FDR $q < 0.05$). GTEx samples were collected at a variety of day times, and previous work demonstrates that they can be ordered to recover circadian cycling genes [31]. Our result suggests that available human DNase profiles fail to recover this variance, but uniquely obtainable mouse time series do.

We hypothesized that the improved accuracy of the jointly trained models on held out chromosome sequences would carry over to more effective variant analysis. To assess this, we computed SLDP Z-scores to GTEx tissues for all human and mouse CAGE and DNase datasets using models trained jointly or on single genomes. For both species, the jointly trained models achieved greater SLDP Z-scores than their single genome counterparts (S11 Fig). Thus, multi-genome training leads to greater concordance between variant effect predictions and GTEx summary statistics.

Finally, we compared our variant effect prediction pipeline to an analogous state of the art pipeline called DeepSEA via SLDP Z-scores versus GTEx summary statistics for all tissues [6]. We computed the difference between the two alleles for the same set of variant using the latest DeepSEA "beluga" model [7]. We manually matched human DNase datasets between the two models, arriving on 100 unambiguous matched datasets. Z-scores from our cross-species Basenji model predictions were greater than those from the DeepSEA model predictions for

69.7% (permuting Basenji/DeepSEA labels p-value $<1 \times 10^{-9}$) (S12 Fig). This result suggests that our regulatory variant annotation pipeline represents the current state of the art among this family of approaches for extracting information relevant to gene expression variation in human populations.

## Mendelian disease variant classification

Having established the relevance and specificity of mouse dataset predictions for expression phenotypes, we asked whether these models could provide insight into the genetic basis of human disease. Mouse data has proven valuable for studying human genetic variants in previous work [20, 25], but these analyses were limited to studying variants in homologous sequences in their mouse genome context. The substantial regulatory sequence turnover between these genomes makes this limitation severe. Our predictive framework avoids this limitation by projecting the learned mouse regulatory grammar to the human genome setting for all variants via machine learning.

We assembled a set of 660 validated noncoding variants implicated in human disease from the HGMD and ClinVar databases that are $> 20$ bp away from mRNA splice sites (Methods) [32, 33]. We constructed a set of negative variants that have matching nucleotide composition, with each variant located approximately 1000 bp from a pathogenic variant to control for genomic region (Methods). We trained random forest classifiers to distinguish variants from these two classes based on features derived from various sets of variant effect predictions in eight fold cross validation. We selected the default hyper-parameters of the scikit-learn 0.22 implementation after finding negligible accuracy gains were available by modifying them [34]. However, due to the large number of features derived from the training datasets, setting the maximum features considered per decision tree split to $log_2$ of the total number of features greatly improved the computational efficiency.

We compared ROC curves produced by models trained on variant predictions for only human datasets (trained jointly with mouse), only human datasets (trained alone without mouse), only mouse datasets, and both human and mouse datasets (Fig 5a). To stabilize accuracy statistics, we repeated the cross validations and stochastic model fitting for 200 iterations and compared models with Mann-Whitney U tests. Human predictions from the model trained jointly produced more accurate classifiers than human predictions from the model trained alone, 0.817 to 0.810 AUROC (p-value $2 \times 10^{-10}$). Adding the mouse predictions as features further improved accuracy to 0.821 (p-value $3 \times 10^{-5}$). Thus, these results suggest that both the joint training procedure and addition of mouse dataset prediction features deliver value for classifying pathogenic noncoding variants. We provide a final random forest classifier trained on the entire variant set with the human and mouse features via https://github.com/calico/basenji for other groups to apply and make use of.

## Complex traits GWAS variant classification

We further hypothesized that these variant effect predictions would deliver insight into noncoding variants highlighted by genome-wide association studies (GWAS) of complex traits and common diseases. We focused our analysis on fine-mapped associations for 47 phenotypes (including 14 common diseases) measured in the UK BioBank [36, 37]. 1524 variants had posterior probability $> 0.95$ of causally affecting one of the phenotypes. We sampled an equal number of negative set variants from marginally associated variants that were determined to have low causal probability $0.001 - 0.01$ in the same analysis. As above, we fit random forest classifiers on variant feature sets derived from various sets of models and predictions.

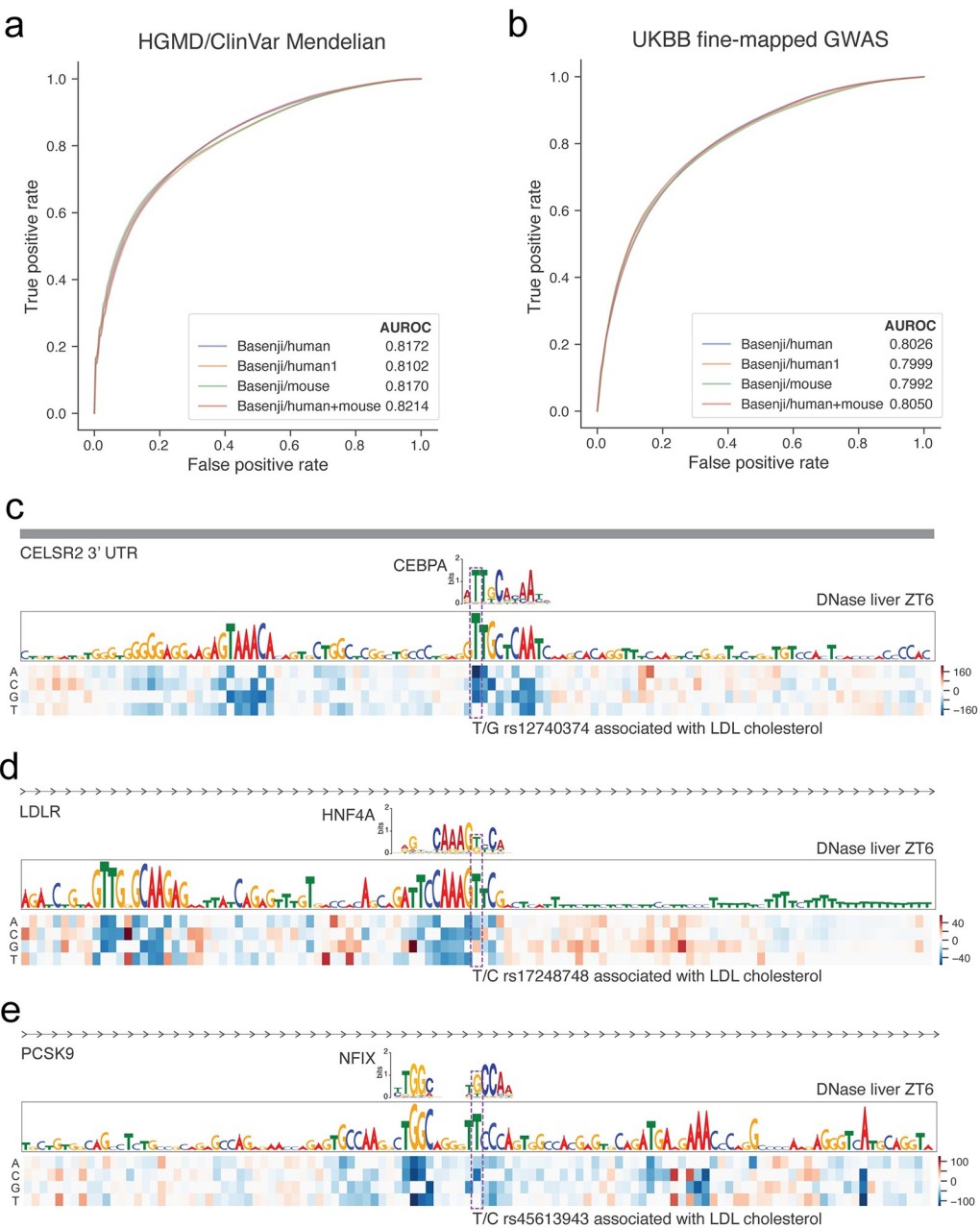

**Fig 5. Multi-species predictions improve variant pathogenicity classification.** (a) The line plots display ROC curves derived from classifiers trained to predict 660 validated noncoding pathogenic variants curated from the HGMD and ClinVar databases from a negative set chosen to control for nucleotide composition and genomic region. "Basenji/human1" uses variant features produced by a model trained on human only, while all other versions use a model trained jointly on both human and mouse. Using this jointly trained model, "Basenji/human", "Basenji/mouse", and "Basenji/human+mouse" produce variant features from predictions for human datasets, mouse datasets, and both human and mouse datasets respectively. For each feature set, we trained random forest classifiers in 200 iterations of eight fold cross validation. (b) We performed an analogous exercise using a set of 1524 variants fine-mapped to have high causal probability > 0.95 for complex traits in the UK BioBank relative to variants with fine-mapped causal probability 0.001 − 0.01. Below, we display three example variants causally implicated to affect LDL cholesterol levels that have large Basenji scores for DNase-seq over a 24 hour time course in mouse liver. (c) rs12740374 creates a CEBPA binding motif in the 3' UTR of CELSR2, and has been experimentally validated to increased liver expression of SORT1 to alter plasma LDL [35]. (d) rs17248748 creates an HNF4A binding motif in in the first intron 6 kb from the TSS of low-density lipoprotein receptor LDLR. (e) rs45613943 breaks an NFI family binding motif in an intron 13 kb downstream of the TSS of PCKS9. Coding mutations in LDLR and PCSK9 have been extensively studied in Mendelian hypercholesterolemia.

As with Mendelian disease variants, fine-mapped GWAS variants can be classified far beyond random guessing. Similar to above, we repeated 200 eight fold cross validation iterations of stochastic model fitting and compared models with Mann-Whitney U tests. Human predictions from a model trained jointly on human and mouse exceeded the accuracy of human predictions from a model trained alone (0.803 versus 0.800 AUROC, p-value $7 \times 10^{-4}$). Adding mouse predictions further improved AUROC to 0.805 (p-value $2 \times 10^{-3}$).

Although recently fine-mapped, most of these variants remain poorly understood. Given the insight above that mouse circadian liver profiles are informative above and beyond human datasets for GTEx liver summary statistics, we focused on variants associated with the related trait LDL cholesterol levels. To reveal regulatory context around these high causal probability variants, we performed saturation mutagenesis of their surrounding region.

rs12740374 emerged as a clear top hit, creating a CEBPA binding motif in the 3' UTR of CELSR2 (Fig 5c). GTEx liver statistics indicate that individuals with the minor allele T have increased expression of nearby genes CELSR2, SORT1, PSRC1, and ATXN7L2 [29]. Musunuru et al. validated the differential CEBPA binding and established that increased liver expression of SORT1 alters plasma LDL [35]. Thus, this variant offers a strong positive control for our analysis pipeline.

We identified several other compelling variants. rs17248748 is located in the first intron 6 kb from the TSS of low-density lipoprotein receptor LDLR. The variant has minor allele frequency 0.5% in the 1000 Genomes reference panel [38], which is too rare to be thoroughly assessed by GTEx as an eQTL. The minor allele T creates a motif that matches the binding preferences of the TF HNF4A, which is a major regulator of liver development and circadian rhythms [39, 40]. Coding variants in LDLR are known to cause Mendelian hypercholesterolemia, but noncoding variants have been far less studied [41]. Another hit, rs45613943, is located in an intron 13 kb downstream of the TSS of PCSK9. The minor allele C has 5% frequency in 1000 Genomes, and GTEx identified its association with decreased PCSK9 expression in whole blood [29, 38]. The reference allele T forms a motif for the nuclear factor I family, which are general activating TFs implicated in a variety of cellular processes. Coding variants in PCSK9 also cause hypercholesterolemia, and their association with LDL has been thoroughly studied and confirmed by the success of PCSK9 inhibitor therapies to lower cholesterol [41]. Together, these variants represent intriguing potential routes by which regulatory variation around critical Mendelian genes modulates their expression to more subtly influence related phenotypes.

## Mouse-trained models highlight mutations relevant to human neurodevelopmental disease

Given the thorough developmental profiling in mouse, we hypothesized that these data might also be a useful lens through which to view human developmental disorders. We retrieved a recent dataset of 1902 quartet families from the Simons Simplex Collection with whole genome sequencing of a mother, father, child affected by autism, and unaffected sibling [42]. In these data, the offspring have an average of 67 de novo mutations, which have a slight enrichment in promoters [43]. Recent work demonstrated that variant effect predictions further differentiate autism probands from their unaffected sibling controls [44]. We hypothesized that predictions using models trained on mouse data would also distinguish the disease and perhaps provide additional insight via novel developmental profiles.

We applied the model to predict how each de novo mutation would influence signal in 357 mouse CAGE profiles of tissues and cell types. We filtered for variants within 50 kb of a GEN-CODE mRNA TSS, where we observed there to be a greater difference between probands and

siblings (S13 Fig), in line with previous analyses [43, 44]. We also observed there to be more signal present in negative variant effect predictions, which indicate mutations that disturb active regulatory elements, and scaled these scores by 10x before taking their absolute value (S14 Fig). Mann-Whitney U (MWU) tests revealed significantly greater variant effect predictions in the probands versus sibling variant sets for 333 CAGE profiles at FDR < 0.05 (Fig 6a, S2 Table). Variant effect predictions did not depend on the age of either parent (S15 Fig). The distribution of P-values for 13 datasets profiling whole body from embryonic and neonate developmental stages was less than the distribution of p-values for the other datasets (MWU p-value $7.7 \times 10^{-4}$). These results were robust to the gene distance filter and negative variant effect prediction weight in a large range (S13 and S14 Figs), but statistical significance was markedly reduced without any gene distance filter. These experiments show all parameter combinations tested, but p-value magnitudes should be interpreted cautiously given the challenge of multiple hypothesis correction in an exploratory analysis with hyperparameters. Although many brain datasets were statistically significant by this test, these data did not collectively stand out among the other CAGE profiles.

We examined the regions around variants predicted to have high impact. For example, a proband variant upstream of *ZNF644* modifies a critical nucleotide in a consensus motif for the transcription factor YY1, which the model identifies as active and relevant (Fig 6b). *ZNF644* has considerable evidence for intolerance to loss of function mutations in the Genome Aggregation Database v2.1.1 (gnomAD) with probability 0.999 of intolerance [45]. YY1 has been implicated in processes that determine the three-dimensional positioning of promoters and enhancers [46]. Thus, we hypothesize that the variant modifies the enhancer regulation of this critical protein.

We observed similar results in Zhou et al.'s independent processing of these raw sequencing data, which resulted in nearly two fold fewer de novo variant calls due to different filters (S16 Fig) [44]. For these variants, 269 CAGE datasets were significant at FDR 0.05, whole body embryonic and neonate developmental stages had lesser p-values (MWU p-value $4.5 \times 10^{-3}$), and brain datasets did not collectively emerge. For additional perspective, we also analyzed DNase and activating histone modifications H3K4me3, H3K4me1, and H3K27ac. For these data, variant effect predictions tended to be larger for proband variants, but the minimum q-value after Benjamini-Hochberg correction of all datasets together was 0.14 (S16 Fig). However, both the set of 259 early development profiles and the set of 144 brain profiles achieved lesser p-values than the rest (MWU p-value $7.7 \times 10^{-4}$ and $1.4 \times 10^{-12}$ respectively). The reduced signal in these data is likely attributable to CAGE's more direct measurement of gene product. The Basenji model has no knowledge of coding sequence or ability to make predictions about coding influence. Nevertheless, we verified that variants overlapping coding sequence evaluated for their influence on gene regulation minimally affect the observations above (S17 Fig). Finally, we also examined human predictions for context, and observed many significant CAGE datasets after FDR correction at 0.05, but no trends for tissue or developmental-specific sets (S18 Fig). Thus, as supported by the conditional SLDP regression to GTEx, the mouse developmental profiles appear to deliver novel information to this analysis.

These significant enrichments indicate that regulatory variant effect predictions may help classify disease at the individual level. For each individual, we computed a simple risk score by summing predictions for a leading developmental dataset mouse CAGE whole body embryonic stage E16. This score suggests more deleterious de novo variants for 54.7% of the probands versus their sibling controls (binomial test p-value $2 \times 10^{-5}$) (Fig 6c). Thus, this approach is a strong candidate for inclusion with complementary feature sources from coding mutations and structural variation to continue to characterize this incompletely understood disorder.

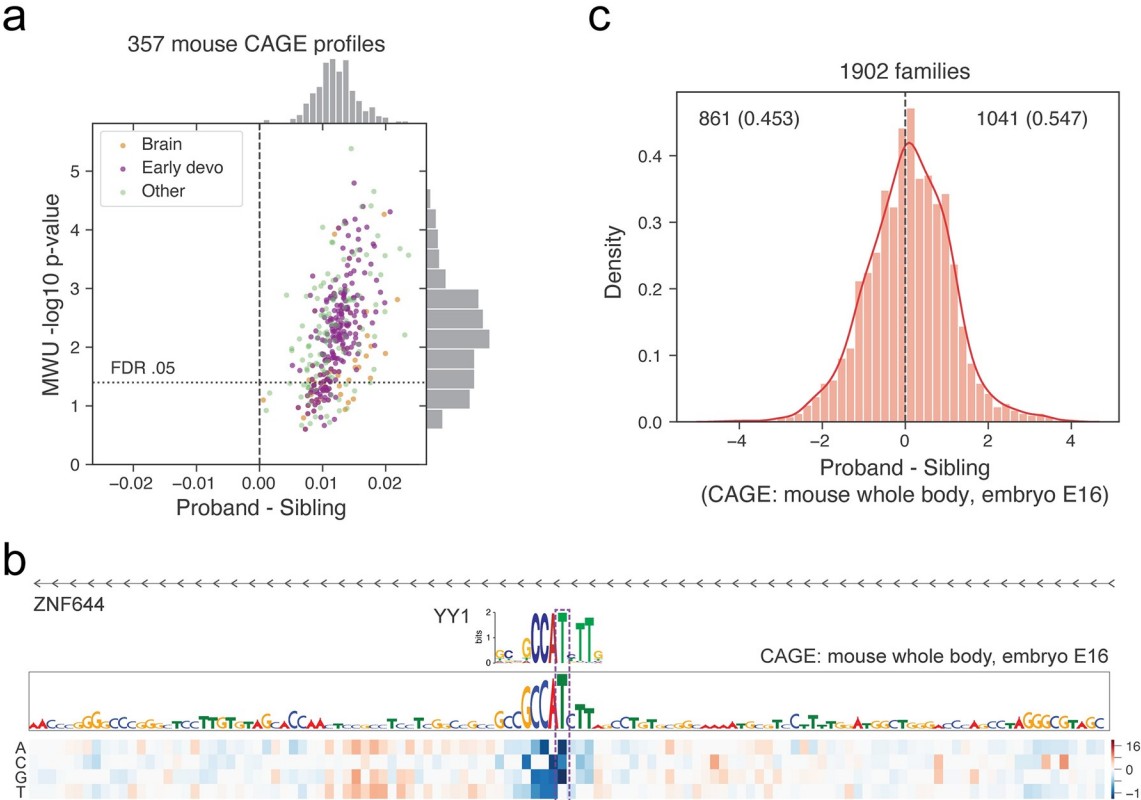

**Fig 6. Human de novo variant predictions for mouse data enrich for autism probands versus their siblings.** (a) We predicted the influence of 234k de novo variants split between probands and sibling controls on 357 CAGE datasets in mouse. For each dataset, we computed a Mann-Whitney U (MWU) test between proband and sibling sets and corrected for multiple hypotheses using the Benjamini-Hochberg procedure. Predictions for many datasets were enriched for greater values in the probands, driven largely by early developmental profiles. Each dataset's x-axis position is the mean natural log over proband variants minus the equivalent over control variants. (b) A proband variant at chr1:91021795 modifies a critical T in a YY1 motif to an A in the promoter region of *ZNF644*. (c) At the individual level, a simple score summing variant predictions for a leading developmental dataset describing mouse CAGE whole body at embryonic stage E16 significantly separates probands from their matched sibling controls (binomial test p-value $2 \times 10^{-5}$).

## Discussion

In this work, we developed a multiple species training procedure to enable a deep convolutional neural network to train multi-task on 6956 functional genomics signal tracks annotating the human and mouse genomes. We observed that training jointly on both species produced models that make more accurate predictions on unseen test sequences relative to models trained on a single species. Regulatory sequence activity predictions for human sequences in mouse tissues correlate well with datasets describing the corresponding human tissues. Model predictions for altered regulatory activity of human genetic variants made with respect to mouse datasets have a strong statistical concordance with tissue-specific human eQTL measurements. Mouse machine learning models can be used to study human disease, exemplified by the beneficial addition of prediction features to disease variant classifiers.

We focused here on human and mouse because both species have been comprehensively studied with genome-wide functional genomics. Our observation that joint training on these two genomes improves prediction accuracy opens the possibility of more complex schemes for training on larger numbers of genomes. Given the substantial evolutionary distance between human and mouse, regulatory annotations for all mammalian genomes are likely to provide

similarly useful training data. Primate genomes will be particularly interesting to explore; their tissues and cell types will more closely match those of human, but their highly similar sequences deliver less novelty. Prediction accuracy improved more for CAGE gene expression measurements than accessibility or ChIP-seq, which suggests that multiple genome training will be worthwhile for data with high regulatory complexity and distal interactions. Efforts to predict spatial contacts between chromosomes as mapped by Hi-C and its relatives fit this criteria, and we hypothesize that training sequence-based models on human and mouse data together will be fruitful [47].

Much prior work has revealed the similarity of regulatory grammars across species, but transferring knowledge gleaned from an accessible model organism (such as mouse) to another of interest (such as human) has remained challenging. Many existing approaches rely on whole genome alignments to transfer annotations from one genome to the other [25, 48]. These approaches are constrained by the quality of the alignment, which is a notoriously challenging bioinformatics problem [49], and the limited proportion of each genome that aligns (40% for human and 45% for mouse). Here, we demonstrated an alternative approach, in which a machine learning model trained on the model organism data compresses the relevant knowledge into its parameters. The model can then be applied to predict activity for sequences from a different genome of interest. The strong tissue-specific statistical relationship between human genetic variant predictions from model parameters trained to predict mouse annotations and GTEx tissue-specific eQTLs highlights the successful nucleotide resolution of our cross-species transfer of learned regulatory grammars. The Gene Expression Omnibus (GEO) contains tens of thousands of mouse functional genomics profiles, many describing experiments impossible in humans. For example, we included dozens of datasets describing mouse liver profiles over 24 hour time courses to study the circadian rhythms of gene expression and chromatin. Models trained to predict the data studied here, as well as open source software to compute these predictions and train new models on users' own data, are available in the Basenji software package [50].

## Materials and methods

### Functional genomics data

In this work, we studied quantitative sequencing assays performed on human and mouse samples. Specifically, we focused on DNase and ATAC-seq profiling DNA accessibility, ChIP-seq profiling TF binding or histone modifications, and CAGE profiling RNA abundance derived from 5' transcription start sites. Preprocessing these data effectively is critical to successful machine learning. Our primary preprocessing objective is to denoise the raw data to the relevant signal at fine resolution.

We largely followed the preprocessing pipeline described in prior research introducing the Basenji framework [5]. The standard pipeline through which experimental data flowed follows:

1. Trim raw sequencing reads using fastp, which can automatically detect and remove unwanted adapter nucleotides [51].

2. Align reads using BWA to hg38 or mm10 and requesting 16 multi-mapping read positions [52].

3. Estimate nucleotide-resolution signal using an open source script (*bam_cov.py*) from the Basenji software that distributes multi-mapping reads using expectation maximization, normalizes for GC bias, and smooths across positions using a Gaussian filter [5].

However, we diverted from this standard pipeline for all data available from the ENCODE consortium website, which is 4,506 human and 1,019 mouse experiments. These data have been thoughtfully processed using open source pipelines and are available for download at several stages, including log fold change signal tracks in BigWig format [53]. Rather than reprocess these data without full knowledge of how replicate and control experiments match, we chose to use these signal tracks directly. The Seattle Organismal Molecular Atlas (SOMA) server provides a single cell mouse ATAC-seq atlas [25]. These data are also available in log fold change BigWig format, and we similarly chose to use these rather than reprocess the single cell data. We clipped negative values in all such BigWig tracks to zero.

We applied several transformations to these tracks to protect the training procedure from large incorrect values. First, we collected blacklist regions from ENCODE and added all RepeatMasker satellite repeats [54], which we found to frequently collect large false positive signal [55]. We further defined unmappable regions of >32 bp where 24-mers align to >10 genomic sites using Umap mappability tracks [56]. We set signal values overlapping these regions to the $25^{th}$ percentile value of each dataset. Finally, we soft clipped high values with the function $f(x) = min(x, t_c + sqrt(max(0, x - t_c)))$. Above the threshold $t_c$ (chosen separately for each experiment and source), this function includes only the square root of the residual $x - t_c$ rather than the full difference. We manually chose $t_c$ per experiment and source by inspecting the maximum values, aiming to reduce the contribution of rare very large values that one would not expect to generalize to other genomic locations. Via this procedure, we decided to clip all CAGE data with $t_c = 384$, ENCODE with $t_c = 32$, and GEO with $t_c = 64$.

When replicate experiments profiling the same or related samples were available, we averaged the signal tracks. Altogether, the training data includes 638 CAGE, 684 DNase/ATAC, and 3991 ChIP datasets in human and 357 CAGE, 228 DNase/ATAC, and 1058 ChIP datasets in mouse. S1 Table describes all data with preprocessing parameters. Code to preprocess typical functional genomics data formats into TensorFlow input formats is available from https://github.com/calico/basenji.

## Model architecture

We modeled genomic regulatory sequence activity signal as a function of solely DNA sequence using a convolutional neural network. Such deep learning architectures have excelled for many similar tasks [2, 4–6]. We follow our prior work in analyzing large 131 kbp sequences in order to consider long range interactions.

The first stage of the architecture aims to extract the relevant sequence motifs from the DNA sequence using the following block of operations:

1. Convolution width 5 (or 15 in first layer)

2. Batch normalization

3. Gaussian Error Linear Unit (GELU) activation

4. Max pool width 2

We applied this block seven times so that each sequence position represents 128 bp, increasing the number of filters from an initial 288 by 1.1776x each block to 768 filters by the end. The GELU activation slightly outperformed the more common ReLU in our benchmarks [57].

The second stage of the architecture aims to spread information across the sequence to model long range interactions. In prior work, we applied densely connected dilated convolutions for this task [5]. Here, we applied a related but more effective variation, which is related to a strategy applied for DNA sequence analysis in the SpliceAI system [58]. Recent deep

learning research has revealed that skip connections between layers where one layer's representation is directly added to a subsequent layer's representation relieve vanishing gradients and improve gradient descent training [21]. Thus, we applied the following series of operations:

1. GELU activation

2. Dilated convolution width 3, dilation rate *d*, 384 filters

3. Batch normalization

4. GELU activation

5. Convolution width 1, back to 768 filters

6. Batch normalization

7. Dropout probability 0.3

8. Addition with the block input representation before step 1.

We applied this block eleven times, increasing the dilation rate *d* by 1.5x each time. Relative to the densely connected version, the dilated residual blocks lead to improved generalization accuracy (S1 Fig).

In the final stage, we first transformed this 1024x768 (length x filters) representation of 128 bp windows with an additional width 1 convolution block using 1536 filters and dropout probability 0.05. To make predictions for either 5313 human or 1643 mouse datasets, we applied a final width one convolution followed by a softplus activation ($f(x) = log(1 + e^x)$) to make all predictions positive. We attached a genome indicator bit to each sequence to determine which final layer to apply.

We trained to minimize a Poisson log likelihood in the center 896 windows, ignoring the far sides where context beyond the sequence is missing. The Poisson model is not technically appropriate for the log fold change tracks. However, by clipping negative values to zero, the distribution of values resembles that from our standard processing. Clipping to zero focuses attention on signal magnitude in regions where relevant signal is present and away from less relevant signal fluctuations in background regions. On a subset of data, we observed that using the log fold change track did not decrease accuracy or the utility of the model for genetic variant analysis.

We implemented the network in TensorFlow and used automatic differentiation to compute gradients via back propagation [59]. We minimized with stochastic gradient descent (SGD) on batches of 4 sequences. We stopped training when the loss on a validation set had not improved for 30 epochs and returned to the model weights that had achieved the minimum validation loss. We performed several grid searches to choose model and optimization hyper parameters for the following sets: (1) SGD learning rate and momentum; (2) initial convolution filters and convolution filter scaling rate; (3) dilated convolution filters and dropout rate; (4) final convolution filters and dropout rate.

Data augmentation describes a suite of techniques to expand the implicit size of the training dataset from the perspective of model training by applying transformations that preserve annotations to data examples. We tiled the 131,072 bp sequences across the chromosomes by 65,599 bp, representing a 50% overlap minus 63 bp in order to also shift the 128 window boundaries and max pooling boundaries. During training, we cycled over combinations of two transformations that maintain the relationship between sequence and regulatory signal while changing the model input: (1) reverse complementing the sequence and reversing the signal;

(2) shifting the sequence 1-3 bp left or right. Both transformations improved test accuracy and reduce overfitting in our benchmarks.

Model implementations and instructions for re-training, predicting, and modifying them are available from https://github.com/calico/basenji.

## Multi-genome training

Training on multiple genomes containing orthologous sequence complicates construction of holdout sets. Independently splitting each genome's sequences would allow training on a human promoter and testing on its mouse orthologue. If the model memorized conserved elements of the sequence, rather than learning a general function, we might overestimate generalization accuracy.

We used the following procedure to minimize occurrence of this potential issue:

1. Divide each genome into 1 mb regions.

2. Construct a bipartite graph where vertexes represent these regions. Place edges between two regions if they have >100 kb of aligning sequence in a whole genome alignment.

3. Find connected components in the bipartite graph.

4. Partition the connected components into training, validation, and test sets.

We used the hg38-mm10 syntenic net format alignment downloaded from the UCSC Genome Browser site [60]. Using this procedure, we set aside approximately 12% of each genome into validation and test sets respectively. Stricter parameter settings created a single large connected component that did not allow for setting aside enough validation and test sequences. We estimated that < 1% of the validation and training set nucleotides have orthologous sequence from the other genome in the training set.

Another complication of training on multiple genomes arises from imbalance between each genome's sequences and datasets. We extracted 38.2k human and 33.5k mouse sequences for analysis. We assembled batches of sequences from one genome or the other, chosen randomly proportional to the number of sequences from each genome. The overall loss function comprises a term for every target dataset summed, which leads to larger step magnitudes for batches of human sequences that are annotated with >3 times more datasets. Explicit weighting could be applied to preference training towards a particular species, but we found this to be unnecessary in our experiments for good mouse performance.

Jointly training on both human and mouse data constrains the model slightly more than is ideal. We found that training on only one genome or the other after the full joint procedure improved validation set accuracy. We evaluated the model on the validation set after every epoch and stopped training after 10 epochs without improvement, returning to the previous model that achieved the minimum validation loss. The model studied here was fine-tuned for 8 epochs on the human data and 20 epochs on the mouse data.

## GTEx SLDP

We predicted the effect of a genetic variant on various annotations by computing a forward pass through the convolutional network using the reference and alternative alleles, subtracting their difference, and summing across the sequence to obtain a single signed score for each annotation. We averaged scores computed using the forward and reverse complement sequence and small sequence shifts to the left and right. We computed scores for all 1000 Genomes SNPs, which we provide for download via instructions at https://github.com/calico/basenji/tree/master/manuscripts/cross2020.

Signed linkage disequilibrium profile (SLDP) regression is a technique for measuring the statistical concordance between a signed variant annotation $v$ and a genome-wide association study's marginal correlations $\hat{\alpha}$ between variants and a phenotype [8]. The functional correlation between $v$ and the true variant effects on the phenotype describes how relevant the annotation is for the phenotype's heritability. Our model produces these signed variant annotations. SLDP estimates this functional correlation using a generalized least-squares regression, accounting for the population LD structure. It performs a statistical test for significance by randomly flipping the the signs of entries in $v$ in large consecutive blocks to obtain a null distribution. We follow previous work in conditioning on background annotations describing minor allele frequency and binary variables for variant overlap with coding sequence (and 500 bp extension), 5' UTR (and 500 bp extension), 3' UTR (and 500 bp extension), and introns.

We downloaded GTEx v7a summary statistics for 48 tissues [29]. We summarized each SNP's effect on all cis-genes using the following transformation suggested for SLDP analysis

$$\hat{\alpha_m} = \frac{1}{\sqrt{|G_m|}} \sum_{k \in G_m} \hat{\alpha}_m^{(k)}$$

where $G_m$ is the set of all genes for which a cis-eQTL test was performed for variant $m$ and $\hat{\alpha}_m^{(k)}$ is the marginal correlation of SNP $m$ and gene $k$ expression [8]. We passed $\hat{\alpha_m}$ to SLDP for analysis of variant predictions.

To assess the orthogonal value of prediction scores derived from mouse datasets relative to those from human, we added the human dataset predictions to the background annotation set. Conditioning on thousands of annotations was computationally intractable. Instead, we included 64 principal components of the human variant scores matrix, which explained $> 99\%$ of the variance in all cases studied. To assess statistical significance, we performed the Benjamini-Hochberg procedure to correct for multiple hypotheses within each GTEx tissue.

## ClinVar and Human Gene Mutation Database

We acquired a set of 15,751 pathogenic noncoding variants curated by di Iulio et al. from ClinVar and the Human Gene Mutation Database (HGMD) [32, 33, 61]. These variants were enriched near splice sites, and many presumably influence splicing. As we do not position our method to predict splicing effects, we removed all variants within 20 bp of GENCODE v28 mRNA splice sites. We further noticed that the remaining variants often clustered together in the genome. To establish a set of variants that function independently, we grouped variants within 10 bp into sets and sampled one representative variant from each set. After applying these filters, 660 validated noncoding variants remained.

We sought a negative set of variants for training machine learning classifiers that roughly controlled for genomic region and nucleotide composition. Using the following procedure, we selected a negative example corresponding to each pathogenic variant. First, we considered positions shifted up and downstream in the genome by 1000 bp. From these anchor positions, we located the nearest reference nucleotide that matched the reference nucleotide of the pathogenic variant. We chose the shifted position that was closer to 1000 bp from the pathogenic variant position and broke ties randomly. Finally, we chose the alternative allele to match the pathogenic variant. Thus, each negative variant matches the nucleotides of a pathogenic variant, while being shifted as close to 1000 bp as possible.

### Simons Simplex Collection

We downloaded 255,106 de novo variants derived from whole-genome sequencing of 1,902 quartet families with an autistic child from the Simons Simplex Collection from the supplement of [43]. For validation, we further downloaded 127,140 de novo variants derived from 1,790 families from the same collection processed by Zhou et al [44]. We filtered all variants for SNPs (i.e. removed insertions and deletions) and computed predictions as described above. We focused our analysis on variants within 50 kb of TSS defined by GENCODE v28. All variant scores are available from https://console.cloud.google.com/storage/browser/basenji_barnyard/sad/autism/.

## Supporting information

**S1 Fig. Dilated residual layers improve accuracy.** We trained two separate models with approximately matched parameter totals on both human and mouse data jointly. The two models differ in how their dilated convolution layers are connected. In the first model *Dense*, which achieved the previous best for these data in [5], each layer takes all previous dilated layers as input, as opposed to taking only the preceding layer. In the second model *Residual*, introduced here, each layer takes only the previous layer as input, transforms it, and adds the new representation into the input before passing on. For each model, we computed the Pearson correlation of test set predictions and observed experimental data for thousands of datasets from various experiment types. (a) Training and validation loss curves for two replicates of the two architectures with random initializations and shuffled training examples. The top curves represent the training set and the bottom curves represent the validation set. Validation losses are less than training losses due only to stochasticity in the sequence splitting procedure. (b) Human and (c) mouse test set Pearson correlation for the best *Dense* versus *Residual* model. (TIF)

**S2 Fig. Multi-genome model assigns greater saliency scores to distal TSS regions.** For 3,523 gene transcription start sites (TSS) that were not included in the training set, we computed saliency maps for the surrounding region using the model trained jointly on human and mouse (joint) and the model trained on human alone (single). The saliency map scores annotate 128 bp segments with a function of the model predictions' gradient with respect to that segment's vector representation after the convolutional layers and before the dilated convolutions share information across wider distances [5]. Peaks in this saliency score detect distal regulatory elements, and its sign indicates enhancing (+) versus repressing (−) influence. (a) For each 128-bp segment, we computed the mean score across genes for liver CAGE. Patterns were consistent across CAGE datasets. (b) For segments greater than 10 kb from the TSS, the mean multi-genome model scores are greater than their single genome counterparts. This suggests that distal enhancer elements are more effectively used to predict gene expression.
(TIF)

**S3 Fig. Multi-genome training accuracy across experiment types.** We trained the same architecture on human data alone and both human and mouse data jointly. For each model, we computed the Pearson correlation of test set predictions and observed experimental data for human CAGE, DNase, and ChIP-seq datasets. Points in the scatter plots represent individual datasets, with single genome training accuracy on the x-axis and joint training accuracy on the y-axis. We considered these accuracy comparisons, broken down by experiment class for the 24 most frequent experiments. Within the various categories, the improvement differed slightly. For example, H3K4me3 increased by 0.006, but H3K9me3 decreased by 0.007.

Enhancer marks H3K4me1 (-0.004), H3K27ac (0), and P300 (-0.001) were stable or unimproved.
(TIF)

**S4 Fig. Multi-genome training accuracy across DNase organ systems.** We trained the same architecture on human data alone and both human and mouse data jointly. For each model, we computed the Pearson correlation of test set predictions and observed experimental data for human DNase datasets. Points in the scatter plots represent individual datasets, with single genome training accuracy on the x-axis and joint training accuracy on the y-axis. We considered these accuracy comparisons in the context of fifteen organ system annotations assigned by Meuleman et al. [26]. Pearson R improves by an average of 0.006 across all of these DNase datasets. Within the various categories the improvement differed slightly. "Musculoskeletal" datasets improved by 0.008, which was significantly greater than the remainder by Mann-Whitney U test with p-value $2e - 7$. In contrast, "Connective" datasets improved by 0.004, which was significantly less than the remainder with p-value $1e - 10$. We were unable to discern a pattern that provided insight into why some categories improve more or less than others with joint training.
(TIF)

**S5 Fig. Multi-genome training harms generalization accuracy for some human ChIP-seq datasets.** To further explore human ChIP-seq datasets that performed worse during multi-genome training relative to single genome training, we trained two independent replicates for both training modes. For the single genome training, we took the average of the two replicates. (a) For the multiple genome training, we plot the ChIP-seq test set PearsonR for replicate 1 minus the single genome PearsonR versus that for replicate 2. Dataset accuracy was consistent across replicates, and some ChIP-seq datasets consistently achieved lower accuracy after multi-genome training. (b) The table displays the 20 datasets with the largest decrease in test set accuracy after multi-genome training. Datasets describing (a) ATF4/CREB binding (known co-factors), (b) K562 cells, and (c) MCF-7 cells performed significantly worse according to Mann-Whitney U comparisons of the sets of 16, 476, and 129 datasets respectively. Histograms consider the average of test set PearsonR for multi-genome training minus the average for single genome training.
(TIF)

**S6 Fig. Additional human datasets do not improve generalization accuracy.** We designed several experiments to explore whether more datasets alone improve the models, rather than the datasets and novel sequence offered by the mouse genome. We split all human datasets into eight, four, three, or two folds. For each fold, we held out those datasets and trained a model only on the remainder. We trained four replicate models from random initializations on the full data for comparison. For the three fold experiment, we repeated the procedure with a unique random split twice. For the two fold experiment, we repeated the procedure with a unique random split three times. For each dataset, we averaged test set accuracy for the training runs that did train on it, each of which had a different portion of the datasets held out. Above, we scatter plot average test set accuracy for each of the 5,313 datasets for the full data versus the held out data runs for each experiment. If it were true that adding more annotations benefited model training and accuracy, then these models would suffer from the held out targets and show reduced accuracy relative to the full model. Instead, for each experiment with hold outs up to half of the datasets, these models achieved slightly greater mean accuracy than the full data model.
(TIF)

**S7 Fig. Cross-species prediction results are consistent across models.** Tissue-specific regulatory programs can be learned and transferred across species, exemplified here by mouse predictions for CAGE (top row) and DNase (bottom row) for cerebellum, liver, and CD4+ T cells. Mouse predictions correspond to mouse datasets matched and compared to human datasets For CAGE, we considered the top 50% most variable TSSs, where data or predictions were quantile normalized to align sample distributions, log transformed, and mean-normalized across samples. For DNase, we considered the top 10% most variable genomic sites (less than CAGE because we consider the whole genome rather than TSSs), where data or predictions were similarly were quantile normalized to align sample distributions and mean-normalized across samples. The statistical trends were robust to most variable threshold choice. Tissue-specific cross-species accuracy depends only slightly whether the mouse model was trained jointly with human data (left column) or alone (right column). This is expected, given that the multi-genome model is more accurate on held out sequences (Fig 2).
(TIF)

**S8 Fig. Cross-species prediction accuracy approaches that of human.** Tissue-specific regulatory programs can be learned and transferred across species, exemplified here by mouse predictions for CAGE (a,b) and DNase (c,d) for cerebellum, liver, and CD4+ T cells. "Human predicted" corresponds to predictions for the human datasets, referred to as "human observed"; "mouse predicted" corresponds to predictions for the matched mouse dataset. For CAGE, we considered the top 50% most variable TSSs, where data or predictions were quantile normalized to align sample distributions, and log transformed. In the right column, we mean-normalized across samples; in the left, we did not. For DNase, we considered the top 10% most variable genomic sites (less than CAGE because we consider the whole genome rather than TSSs), where data or predictions were similarly quantile normalized to align sample distributions and mean-normalized across samples in the right column only. The statistical trends were robust to most variable threshold choice. (a,c) Human prediction accuracies exceed (b,d) mouse prediction accuracies for both CAGE and DNase.
(TIF)

**S9 Fig. Residuals for human and mouse predictions differ with respect to local GC content.** For the DNase sites studied across species, we computed residuals as the mean-normalized observed signal minus predicted signal. We computed GC% in a 1,000 bp region around the 128 bp segment. Correlations between GC content and the residuals were larger for mouse than human, indicating that mouse predictions may be slightly mis-calibrated for the human genome.
(TIF)

**S10 Fig. Variant effect predictions for mouse datasets significantly correlate with GTEx via SLDP, even conditional on human dataset predictions.** We computed variant effect predictions for all 1000 Genomes variants with respect to human and mouse datasets. We then analyzed the CAGE and DNase/ATAC data separately. We computed the first 64 principal components (PCs) of the variants by human predictions matrix, which explained 99.9% of the variance for CAGE datasets and 99.3% for DNase/ATAC. We then computed the statistical correlation between mouse predictions and GTEx summary statistics across 48 tissues using SLDP conditioned on the 64 human PCs for either the CAGE or DNase/ATAC data (Methods). (a) For the tibial artery and (c) left ventricle GTEx summary statistics, mouse CAGE datasets describing the developing heart in neonate and embryo stages emerged as significant after Benjamini–Hochberg correction for multiple hypotheses. Prefix asterisks indicate FDR $q < 0.05$. Additional datasets describing adult heart components and muscle also reach

significance. (b) For liver GTEx, mouse single cell hepatocyte and DNase datasets describing a 24 hour time course to profile circadian rhythms of genome accessibility reach significance. (TIF)

**S11 Fig. Variant effect predictions from jointly trained models correlate better with GTEx via SLDP.** We computed variant effect predictions for all 1000 Genomes variants with respect to human and mouse datasets using models trained jointly on both human and mouse or trained alone on a single genome. We then computed the statistical correlation between these predictions and GTEx summary statistics across 48 tissues using SLDP (Methods). The points underlying the density maps represent every pair of model prediction dataset and GTEx tissue. SLDP signed Z-scores indicate the expected positive statistical relationship between predictions for CAGE, DNase, and ATAC-seq and gene expression. These Z-scores are clearly greater for predictions from jointly trained models for (a) human CAGE, (c) mouse CAGE, and (d) mouse DNAase/ATAC. (b) Human DNase/ATAC Z-scores are more similar between the joint and single trained models, in line with their comparable accuracy on held out sequences (Fig 2). (TIF)

**S12 Fig. Variant effect predictions from cross-species Basenji pipeline correlate better with GTEx via SLDP than DeepSEA pipeline.** We computed variant effect predictions for all 1000 Genomes variants using the DeepSEA "beluga" model. We then computed the statistical correlation between these predictions and GTEx summary statistics across 48 tissues using SLDP (Methods). We manually aligned all human DNase datasets between the DeepSEA and Basenji models, arriving at 100 matched datasets. Here, we scatter plot Basenji versus DeepSEA Z-scores for each combination of DNAse dataset and GTEx tissue. Basenji Z-scores are greater for 69.7% of combinations (permuting Basenji/DeepSEA labels p-value $<1 \times 10^{-9}$). (TIF)

**S13 Fig. Autism de novo variant predictions for mouse CAGE datasets across variant TSS distance filters.** We predicted the influence of de novo variants found in proband and sibling genomes for mouse CAGE datasets. We studied two versions of the processed whole genome sequences by An et al. (left column) and Zhou et al. (right column) [43, 44]. In each plot, points represent CAGE datasets, from which a predicted activity difference can be assigned to each variant. We scaled negative variant scores by 10 before taking the absolute value. In each row, we filtered for variants within the specified distance of a GENOCODE mRNA TSS. On the x-axis, we plot the mean natural log score for all proband variants subtracted by the mean natural log score for all sibling variants. On the y-axis, we plot the $log_{10}$ p-value from comparing proband to sibling scores using a Mann-Whitney U test. Variants nearby TSS contain more signal than those that are very far from genes, which may nonetheless have nonzero CAGE predictions due to enhancer RNA activity. At the most strict distance threshold of 30kb, the x-axis difference between probands and siblings continues to grow, but the y-axis significance is decreased by the smaller number of remaining variants. The Zhou et al. datasets are robust to the filter distance, but the An et al. datasets only reach FDR q-values < 0.1 with TSS filters < 500 kb (which keeps 90% of variants) and decrease as the filter distance tightens around TSS. (TIF)

**S14 Fig. Autism de novo variant predictions for mouse CAGE datasets across negative prediction weights.** We predicted the influence of de novo variants found in proband and sibling genomes for mouse CAGE datasets. We studied two versions of the processed whole genome sequences by An et al. (left column) and Zhou et al. (right column) [43, 44]. In each plot,

points represent CAGE datasets, from which a predicted activity difference can be assigned to each variant. We focused on variants within 50 kb of a GENCODE mRNA TSS. In each row, we scaled negative variant scores by the specified factor, before taking the absolute value. On the x-axis, we plot the mean natural log score for all proband variants subtracted by the mean natural log score for all sibling variants. On the y-axis, we plot the $log_{10}$ p-value from comparing proband to sibling scores using a Mann-Whitney U test. The first row displays the statistical tests computed for only negative values, which represents the limit as the negative prediction weight grows to infinity. Negative predictions are more informative than positive, so an even weighting of the two produces less significant differentiation between the probands and their sibling controls. However, the results are very robust to the choice of the scaling factor.
(TIF)

**S15 Fig. Autism de novo variant predictions do not depend on parent age.** We predicted the influence of de novo variants found in proband and sibling whole genomes sequences processed by An et al. for mouse datasets [43]. Variant predictions are derived from a representative dataset of "whole body, embryo E16" profiled by CAGE that has significantly greater effect score for probands relative to their sibling controls. We transformed the raw variant predictions similarly to the main analyses by scaling negative variant scores by 10 before taking the absolute value and further adding a pseudocount of one and taking the natural logarithm to improve the stability of the scores for the visualization. Regression analysis indicates no dependency of the scores on the mother or father's age at birth. This result matches that of Zhou et al. in their analysis of a separate processing of these data. [44].
(TIF)

**S16 Fig. Autism de novo variant predictions for mouse datasets.** We predicted the influence of de novo variants found in proband and sibling genomes for mouse datasets. We studied two versions of the processed whole genome sequences by An et al. (left column) and Zhou et al. (right column) [43, 44]. We separated CAGE gene expression (top row) from active chromatin modifications DNase/ATAC/H3K4me3/H3K4me1/H3K27ac (bottom row). In each plot, points represent datasets, from which a predicted activity difference can be assigned to each variant. We focused on variants within 50 kb of a GENCODE mRNA TSS and scaled negative variant scores by 10 before taking the absolute value. On the x-axis, we plot the mean natural log score for all proband variants subtracted by the mean natural log score for all sibling variants. On the y-axis, we plot the $log_{10}$ p-value from comparing proband to sibling scores using a Mann-Whitney U test. Many CAGE datasets show Benjamini-Hochberg q-values < 0.05; chromatin datasets demonstrate a similar trend toward greater scores for proband variants, but do not reach the same significance levels.
(TIF)

**S17 Fig. Autism de novo variant predictions for mouse datasets, without coding overlap.** We predicted the influence of de novo variants found in proband and sibling genomes for mouse datasets. We studied two versions of the processed whole genome sequences by An et al. (left column) and Zhou et al. (right column) [43, 44]. We removed any variant overlapping GENCODE coding sequence, 1.7% and 3.3% respectively. We separated CAGE gene expression (top row) from active chromatin modifications DNase/ATAC/H3K4me3/H3K4me1/H3K27ac (bottom row). In each plot, points represent datasets, from which a predicted activity difference can be assigned to each variant. We focused on variants within 50 kb of a GENCODE mRNA TSS and scaled negative variant scores by 10 before taking the absolute value. On the x-axis, we plot the mean natural log score for all proband variants subtracted by

the mean natural log score for all sibling variants. On the y-axis, we plot the $log_{10}$ p-value from comparing proband to sibling scores using a Mann-Whitney U test. Many CAGE datasets show Benjamini-Hochberg q-values < 0.05; chromatin datasets demonstrate a similar trend toward greater scores for proband variants, but do not reach the same significance levels. (TIF)

**S18 Fig. Autism de novo variant predictions for human datasets.** We predicted the influence of de novo variants found in proband and sibling genomes for human datasets. We studied two versions of the processed whole genome sequences by An et al. (left column) and Zhou et al. (right column) [43, 44]. We separated CAGE gene expression (top row) from active chromatin modifications DNase/ATAC/H3K4me3/H3K4me1/H3K27ac (bottom row). In each plot, points represent datasets, from which a predicted activity difference can be assigned to each variant. We focused on variants within 50 kb of a GENCODE mRNA TSS and scaled negative variant scores by 10 before taking the absolute value. On the x-axis, we plot the mean natural log score for all proband variants subtracted by the mean natural log score for all sibling variants. On the y-axis, we plot the $log_{10}$ p-value from comparing proband to sibling scores using a Mann-Whitney U test. Many CAGE datasets show Benjamini-Hochberg q-values < 0.05; chromatin datasets demonstrate a similar trend toward greater scores for proband variants, but do not reach the same significance levels. (TIF)

**S1 Table. Training dataset descriptions.** (XLSX)

**S2 Table. Autism de novo variant enrichment statistics.** (XLSX)

## Acknowledgments

Marta Mele-Messeguer for conversations to conceive the research direction. Jacob Kimmel, Leland Taylor, Geoffrey Fudenberg, Vikram Agarwal, and Han Yuan for valuable manuscript feedback.

## Author Contributions

**Conceptualization:** David R. Kelley.

**Data curation:** David R. Kelley.

**Formal analysis:** David R. Kelley.

**Investigation:** David R. Kelley.

**Methodology:** David R. Kelley.

**Resources:** David R. Kelley.

**Software:** David R. Kelley.

**Validation:** David R. Kelley.

**Visualization:** David R. Kelley.

**Writing – original draft:** David R. Kelley.

**Writing – review & editing:** David R. Kelley.

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
