## [Decision Letter · Decision Letter 0]

10 Nov 2019

Dear Dr. Kelley,

Thank you very much for submitting your manuscript 'Cross-species regulatory sequence activity prediction' for review by PLOS Computational Biology. Your manuscript has been fully evaluated by the PLOS Computational Biology editorial team and in this case also by independent peer reviewers. The reviewers appreciated the attention to an interesting problem, but raised a few major concerns about the manuscript as it currently stands. In particular, multiple reviewers pointed out that the comparisons with previously developed methods and prior results in the literature are not adequate. The novelty of the machine learning framework needs to be clarified in the context of other existing approaches. In addition, more comprehensive evaluation and analyses regarding the cross-species comparison should be added to clearly demonstrate the advances and benefit. While your manuscript cannot be accepted in its present form, we are willing to consider a revised version in which the issues raised by the reviewers have been adequately addressed. We cannot, of course, promise publication at that time.

Your revisions should address the specific points made by each reviewer with detailed point-by-point responses. Please return the revised version within the next 60 days. If you anticipate any delay in its return, we ask that you let us know the expected resubmission date by email at ploscompbiol@plos.org. Revised manuscripts received beyond 60 days may require evaluation and peer review similar to that applied to newly submitted manuscripts.

Sincerely,

Jian Ma

Associate Editor

PLOS Computational Biology

William Noble

Deputy Editor

PLOS Computational Biology

[LINK]

Reviewer's Responses to Questions

**Comments to the Authors:**

**Reviewer #1: **In the manuscript “Cross-species regulatory activity sequence prediction” by David Kelley the author presents modifications and applications of his previously developed deep convolutional neural network based framework, basenji, to predicting regulatory sequence activity when jointly considering human and mouse data. The manuscript evaluates the extent to which predictions within a species can be improved by doing multi-task training across species. The manuscript also evaluates how well models trained to predict mouse activity perform when applied to predict activity in human sequence. Models trained using mouse data are then applied to predict the effect of genetic variants and are then related eQTLs in GTEx and de novo variants from autism whole genome sequencing (WGS) data.

Figure 5a appears to be reporting a major result though I am surprised the text is not highlighting the significance of what is being reported. In Fig. 5a the author is reporting case control differences with multiple of their mouse CAGE predictive models at a p-value better than 10^-6 with at least one model getting a p-value better than 10^-7. In contrast, in two recent high profile papers considering the same autism WGS data, in somewhat analogous analyses (Supplementary Fig. 4 from Zhou et al Nature Genetics, 2019 and Fig 1e from An et al, Science 2018) did not get p-values better than 10^-4 despite conducting an order of magnitude more statistical tests. Zhou et al Nature Genetics, 2019 also used a deep learning sequence based prediction strategy. There are a number of differences in how these analyses were conducted so it is hard from what is presented to know what is driving this large improvement in p-values. (I am assuming here the y-axis in Fig 5a is in base 10 scale though that was not explicitly noted)

Given that this has the potential to be a major result, I think the author should present more analyses and information allowing a reader to better understand the result, its relationship to previously reported results in the literature, and be convinced of the biological significance of the result as opposed to being driven by something technical. I have a number of comments/suggestions in this regard:

i) Can the author explain why the p-values appear to be substantially more significant for the mouse CAGE data in Fig. 5a than for the human CAGE data in Sup. Fig. 8 even though the application of the model is within human? Maybe looking at the correlation of predictions for corresponding brain predictions between human and mouse and/or whether anything can be said about the specific variants better predicted using data from one species or the other would help with this.

ii) Is the author currently including protein coding variants in the p-value calculations? If so what happens to the p-values if protein coding variants are excluded from the analysis? Protein-coding variants are known to already have a strong detectable association with ASD so it would be less interesting if the signal is coming from protein coding variants.

iii) An et al, Science 2018 corrected for parental age in computing p-values. If the author corrected for parental age what would happen to their reported p-values?

iv) Zhou et al, Nature Genetics 2019 used a different set of de novo mutational calls from An et al, Science 2018 and used here, with a notable difference of excluding variant calls overlapping repeat elements. If the author applies their predictions on the same set of de novo mutation calls from Zhou et al what p-values result?

v) Are these strong p-values unique to CAGE data or can they also be seen in chromatin data?

vi) To what extent can the author’s predictions recall case de novo mutations at high precision? This type of evaluation could be more relevant than a p-value across the full distribution since the expectation with de novo mutations is that their contribution is through a smaller number of larger effect variants

vii) The methods states “We filtered these variants for SNPs”. Can the author provide more details of what SNP list was used for filtering, how many variants were filtered, and whether this is causing difference with previous analyses?

viii) The author should provide as a supplementary data or a file on a website that provides their scores for each de novo mutation based on each CAGE dataset

Another major comment I have about this manuscript is that I think the author should evaluate the extent to which improvements for human predictions when training with mouse is occurring because of the diversity of sequence space the mouse provides as motivated in the introduction, or if a similar or better improvement would be seen by using an equivalent number of additional datasets from human. The author only used ENCODE and FANTOM5 data in human so is far from exhausting the available data sets on regulatory activity collected directly in human as there are large datasets from other consortium (e.g. Roadmap Epigenomics, Blueprint) and databases that provide uniform reprocessing of major data repositories (e.g. ChIP-atlas and ReMap). The framework the author is proposing might actually make more sense when the primary interest is in a mammal with far less functional genomics data available than mouse or human.

Additional comments

*There are similarities between what the author is doing here and Chen et al, PloS Comp 2018. The authors cite that manuscript in the results section, but I think the author should acknowledge this prior work and clarify the contribution of this manuscript to prior work up front in the introduction.

* On page 10, the authors say their procedure can suggest high level annotations for 6/9 of the unknown single cell clusters. Is this leading to different/better suggestions than using nearby gene expression of cluster peaks?

* The methods could be more detailed/precise in places. For example, how was the threshold t_c chosen? Another example, it is stated “We found that training several epochs on only one genome or the other after the full joint procedure improved validation and test set accuracy.” How many epochs did the author actually use for this final step and how was it determined?

* It is stated “Stricter parameter settings created a single large connected component that did not allow for setting aside enough validation and test sequences”. This does not directly address whether the current settings are adequate to avoid the memorization issue

* page 12 line 239, “negative predictions” used before explaining “negative predictions” at the bottom of the page

*there are ? in sup. Fig 1 and 2 legends

**Reviewer #2:** The author proposed a deep convolutional neural network based model for cross-species regulatory sequence activity prediction. The author trained the model with both human and mouse genomic data and showed that the jointly trained model achieves better performance as compared to the individually trained model. Next, the experiments were designed to show that the model trained in mouse could help reveal human genetic variant effects. Specifically, the author demonstrates that by training a model with a mouse species-specific data, scATAC-seq, it would help better characterize the effects of variants in human as compared to the model trained on other existing data in the human. This may suggest the potential of this work for generalizing patterns learned from one species to another. However, there are a few major problems with this work.

Major comments:

1. In terms of methodology, it lacks novelty. The strategy is simply concatenating two datasets from different organisms and making sure that homologous sequences appear in the train/test set together. This is hardly a new computational development effort. The author also said that a novel transfer learning approach was developed, but it seems that this is just training the model on mouse dataset and predicting on the human dataset, which is not exactly transfer learning. There are no methods such as domain adaption developed for the specific problem. This is more like a cross-species validation/assessment rather than a transfer learning approach.

2. In the first section of the results, the author compared the performance of a jointly trained model versus the model trained with the independent dataset. However, there are existing methods for cross-species/cross-cell type functional genomic signal prediction. For example, Chen et al. PLOS Computational Biology 2018 (PMID: 30286077) and also Lan et al. Int J Mol Sci. 2019 (PMID: 31336830). Although the setting is not exactly the same because one is binary classification while this one is regression, it is rather straightforward to adapt those methods to the setting in this work. The comparison with existing methods is lacking.

3. For the third subsection in Result. The author showed the importance of the orthogonal information brought in by the mouse scATAC signal by including the principal components of human signals in the background model. The results showed that the mouse signals can be important, but the level of importance is still not clear. It would be interesting to design the experiment when the human signal and mouse scATAC signal are jointly tested. This would help to quantitatively analyze the importance of this orthogonal information by comparing the Z-score from the human signal and mouse signal. If in this test the mouse signal consistently shows up at the top, then it can demonstrate the usefulness of this method to generalize signals from another organism to the human better. I think that this needs to be addressed.

Minor points:

Figure 1,3,4 are barely readable. The font size is really small and thin.

**Reviewer #3:** In this manuscript, the author demonstrates the superior performance of a deep convolutional neural network trained on both human and mouse gene regulatory sequences compared to models trained on data from a single species. Furthermore, the predictions from both the joint model and the mouse model were able to predict the effect of human variants, suggesting the generalization of the approach. In general, the manuscript is well written and easy to follow. The author performed an extensive analysis across a variety of genomic data and demonstrated the possible use of other species data in improving the understanding of human genetic variants. The results presented in the manuscript have many merits and also raise a few questions as follows.

Major comments:

The authors demonstrate that CAGE eRNA profile prediction improves more than prediction of other functional genomics features from ENCODE/Roadmap. Are there substantial differences between functional genomics profiles or cell types of origin in terms of the improvement provided by the method introduced here?

Line 102–108. To me “regulatory programs” implies the particular combinatorial patterns of many regulatory elements that drive specific patterns of gene expression. Thus, in my view, it is not necessarily true that the regulatory programs themselves are conserved, but rather that the components of these programs are conserved. To extend the “program” analogy further: human and mouse may have different regulatory programs, but these programs are written in the same language.

In the last Results section “Mouse-trained models highlight mutations relevant to human neurodevelopmental disease”:

- The predictive power of summing the negative predictions in the neonate cerebellum dataset at predicting cases from controls (Figure 5C) seems quite poor. Is it possible to benchmark this performance with the deep learning model used in Zhou et al. Nature Genetics 2019 (PMID: 31133750)?

- In the cross-specifies prediction, such as Figure S5d, the mouse model’s prediction is worse than human predictions. Have you analyzed the sequences that the mouse model failed to predict? Could it be partially driven by the very different GC content landscape in the mouse genome compared to the human genome?

- The result section emphasizes that the mouse-trained model can be applied to understand human disease. While this point is true, given the abundance of human data and potentially better performance of human model (Figure S8) in this autism cohort, it would be useful to also demonstrate that incorporation of mouse data with the human data helps. More specifically, demonstrate that the human predictions from the human-mouse joint model is better than the human predictions from the human only model.

- As a final general comment, the impact of the manuscript would be stronger if (especially in the applied sections of the Results) the authors could compare to other state-of-the-art machine-learning-based methods for regulatory variant interpretation.

Minor comments:

The Introduction (Lines 25 – 32) does not fully cover the previous literature on the prediction of genomic data using multiple species. in paragraph. The author did mention some previous work in the Results section (Line 114). It would be better to cover this background more fully in the Introduction and to cite other relevant work, such as Cohn et al. 2018 (https://www.biorxiv.org/content/10.1101/264200v2).

Line 20. The author mentioned that “Individual human genomes differ only slightly from each other, so acquiring functional profiles for more humans is unlikely to provide this boost.”

I don’t necessarily agree with the point. Although small percentage of the genome differs between individuals, the total amount genetic variants is large and gives rise to the diversity of human phenotypic traits and diseases. I would acknowledge that multiple different ways of obtain more training data may be helpful.

Line 191 – 210. In Figure S6, is the mouse single cell ATAC-seq dataset used in the SLDP? I didn't see any of the mouse scATAC data sets among the significant ones.

Figure 3c,e: It would be helpful to use a simpler color map with white at 0 and a gradient to a single color to represent positive values and to a different single color for negative values. See: https://www.oreilly.com/library/view/fundamentals-of-data/9781492031079/ch04.html

Line 297–299. This claim is not accurate. This manuscript is not the first to propose that machine learning models can be used to extract relevant features independent of alignability to apply across species. In other words, previous approaches for applying machine learning methods to infer regulatory activity and functional genomics feature profiles across species do not all rely on alignability of the underlying sequences.

**Have all data underlying the figures and results presented in the manuscript been provided?**

Reviewer #1: No: Author should provide prediction values for autism mutations part of Fig 5a analysis

Reviewer #2: Yes

Reviewer #3: Yes

PLOS authors have the option to publish the peer review history of their article (what does this mean?). If published, this will include your full peer review and any attached files.

Reviewer #1: No

Reviewer #2: No

Reviewer #3: No

---

## [Decision Letter · Decision Letter 1]

25 Mar 2020

Dear Dr. Kelley,

Thank you very much for submitting your manuscript "Cross-species regulatory sequence activity prediction" for consideration at PLOS Computational Biology.

As with all papers reviewed by the journal, your manuscript was reviewed by members of the editorial board and by several independent reviewers. In light of the reviews (below this email), we would like to invite the resubmission of a significantly-revised version that takes into account the reviewers' comments. In particular, one of the reviewers felt that the revised version did not adequately address some major concerns raised in the previous round of review. We therefore request that you further revise the manuscript to respond to the reviewer's request. We cannot make any decision about publication until we have seen the revised manuscript and your response to the reviewers' comments. Your revised manuscript is also likely to be sent to reviewers for further evaluation.

Sincerely,

Jian Ma

Deputy Editor

PLOS Computational Biology

William Noble

Deputy Editor

PLOS Computational Biology

Reviewer's Responses to Questions

**Comments to the Authors:**

Reviewer #1: In the revised manuscript of “Cross-species regulatory sequence activity prediction” I felt my previous comments were partly addressed. I still have some comments on the manuscript.

1. I previously raised the point about whether the increased predictive power was due to unique information in mouse sequence data, or whether comparable predictive performance increase could be obtained from incorporating additional human data. I pointed out that there are many additional sources of human data not being used. In response, the author showed there was basically no impact on predictive performances when downsampling the current set of human data to 7/8 the size.

I did not find this analysis fully satisfying for two reasons. One reason is that holding 1/8 of the human data sets out amounts to 664 human data sets, while there was 1,643 mouse data sets used, thus the comparison of value of additional human vs. mouse data is confounded by the number of data sets. The other reason is that the analysis is effectively assuming the current human data being considered is representative of all available human data, which is likely not the case. I expect that when one gets data from an additional source, even if it is from the same species, it will contain more new information compared to data obtained in the same way as the data being currently considered.

2. The procedure for the autism analysis was changed to remove variants not within 50kb of a TSS, and to weigh negative predictions 10 times more than positive predictions. I did not suggest either of those specific changes. While the p-values improved (when everything is compared in log10 space), it is now more difficult to interpret the p-values. The reason is that I assume there was some type of multiple testing going on in selecting these parameters, but that procedure was not described and any additional tests were not reflected in a multiple testing correction, but should be. Also, I think it would be informative to report as was done in the original submission how well the scores does directly without going through this filtering and reweighting. The reported predictive performance at the individual level for the top variant feature might also be inflated by multiple testing issues.

3. Related to the previous point, in the original submission there was emphasis on brain being the important cell/tissue type, but now the emphasis is on earlier development stages and the brain does not standout. It is concerning the biological conclusions are that sensitive to these specific parameter choices. Further raising concern, the text mentions ‘whole body embryo E16’ being the leading dataset, but according to Supplementary Table 2, ‘whole body embryo E16’ is third most significant and ‘urinary bladder, adult’ is actually the most significant.

4. The author asked for clarification on the point about parental age. An et al described in their paper their procedure for controlling for parental age and provide parental age annotations with it. Zhou et al did not correct for parental age, but did show that parental age was not correlated with their score. While the author’s score might not be correlated with age, that was not shown. Zhou et al not seeing a correlation with parental age for their score, does not imply that the authors’ scores is not correlated with parental age, though does make that possibility more likely.

5. On the point about annotating unknown clusters from Cusanovich et al. I don’t think its been shown that the author’s procedure is actually leading to better annotations opposed to being willing to annotate a cluster when there is still more uncertainty. For example for cluster 5.6, a number of different brain regions rank highly in Fig. 4b of the author's manuscript, but from the tissue type annotations of the cluster in Fig 2d of Cusanovich et al, it was already clear this cluster was related to the brain.

6. I felt the added comparison with EIGEN and FunSeq2 is confounding two different questions. One question is whether the features produced by Basenji add value to variant prediction over features considered by other variant prioritization methods. The other question is whether there is an advantage to integrating a set of features in task-specific ways that is optimized for the evaluation of the task. Only the Basenji features were integrated in task-specific ways, but it is possible integrating the features of EIGEN or FunSeq2 in a way that is optimized for the evaluation would have led to even better performance than what is being reported for Basenji features.

Reviewer #2: The author addressed my major concerns on the performance evaluation by including an extra section of the comparison with DeepSEA, another state-of-the-art machine learning-based model for regulatory variant interpretation, which demonstrates the advantage of this approach. The author elaborated on the differences of this work's object with the goal of the existing work and released the noncoding variant prediction results which clarify the contribution of this work. The author did improve the quality of the figures to a certain extend. However, I still think the font size of Figure 1,3,4 is too small. The authors and the editors should work together to ensure the quality of the figure.

Reviewer #3: N/A

**Have all data underlying the figures and results presented in the manuscript been provided?**

Reviewer #1: Yes

Reviewer #2: Yes

Reviewer #3: Yes

PLOS authors have the option to publish the peer review history of their article (what does this mean?). If published, this will include your full peer review and any attached files.

Reviewer #1: No

Reviewer #2: No

Reviewer #3: No
---

## [Decision Letter · Decision Letter 2]

21 May 2020

Dear Dr. Kelley,

Thank you very much for submitting your manuscript "Cross-species regulatory sequence activity prediction" for consideration at PLOS Computational Biology. As with all papers reviewed by the journal, your manuscript was reviewed by members of the editorial board and by independent reviewers. One of the reviewers raised additional major questions on your analysis (see comments below) that we hope you can thoroughly address and clarify. Please prepare a point-by-point response and modify the manuscript accordingly.

Sincerely,

Jian Ma

Deputy Editor

PLOS Computational Biology

William Noble

Deputy Editor

PLOS Computational Biology

[LINK]

Reviewer's Responses to Questions

**Comments to the Authors:**

Reviewer #1: Unless I am misunderstanding something, I believe the author is making a basic error that is causing in part an unsupported biological conclusion to be reached in the Autism analysis.

I previously raised concern that early development was being emphasized as being relatively important, while under a different setting of the parameters it wasn’t. Additionally I was concerned that the emphasis was being placed on early development when ‘urinary bladder, adult’ was the most significant hit for the An et al Mouse data sets.

The author in the text and response is saying the conclusions were robust to two different processings of the data. However, despite what the author is saying in the text and the response, for the Zhou et al data, based on what is in Supplementary Table 2 the p-values actually appear less significant for embryo/neonate CAGE data than other CAGE data.

I computed the median p-values for 218 embryo/neonate data CAGE data sets and 139 non-embryo/neonate CAGE data sets (note the text said it was 219 vs. 138 so there is a discrepancy of one) and obtained the following:

Median Zhou et al embryo/neonate: 0.020

Median Zhou et al not embryo/neonate: 0.009

Median An et al embryo/neonate: 0.002

Median An et al not embryo/neonate: 0.006

Additionally, even for An et al data I don’t think it is accurate to describe embryo/neonate datasets as being ‘prominent among these associations’ as I am not seeing them in the top part of the rankings of the most significant CAGE data.

In terms of the issue I raised of not stating which multiple tests were performed and correcting for it, instead of doing that the author declared the analysis an exploratory analysis. As the author acknowledged in the response the hypotheses selected to test were based on following previous significant analyses on the same data, thus because of that bias it would be challenging to formally correct for the multiple testing anyway. The author also presented a robustness analysis for the parameters, though that analysis was limited in that the selected parameter values were always one of the extreme values in the analysis. Additionally the analysis was shown for only the two slices of the grid of hyper-parameter values that had at least one selected value opposed to the entire grid. The author may want to consider expanding the robustness analysis with these considerations, but this is a minor point.

In terms of the issue of evaluating on additional human data, I agree with the author given the effort that the author says would be involved to investigate it, it is likely not worth the effort.

**Have all data underlying the figures and results presented in the manuscript been provided?**

Reviewer #1: Yes

PLOS authors have the option to publish the peer review history of their article (what does this mean?). If published, this will include your full peer review and any attached files.

Reviewer #1: No
---

## [Decision Letter · Decision Letter 3]

12 Jun 2020

Dear Dr. Kelley,

We are pleased to inform you that your manuscript 'Cross-species regulatory sequence activity prediction' has been provisionally accepted for publication in PLOS Computational Biology pending your response to clarify the additional questions from one of the reviewers. 

Best regards,

Jian Ma

Deputy Editor

PLOS Computational Biology

William Noble

Deputy Editor

PLOS Computational Biology

Reviewer's Responses to Questions

**Comments to the Authors:**

**Reviewer #1**: The author modified the association of embryonic and neonate developmental states from the previous version of the manuscript to just be whole body and now states: “Whole body profiles from embryonic and neonate developmental stages were prominent among these associations.”

I still think the author is making an overstatement in saying whole body embryonic and neonate developmental is prominent. For example based on the An et al processing only two of the 13 whole embryo and neonate development stages were in the top 20 and based on the Zhou et al only one was in the top 20. The author does report there is a statistically significant difference in p-values based on a mann whitney test, but I think the word ‘prominent’ implies something stronger. Additionally the statistical significance is based on assuming all the observations are independent, which is debatable since there is likely a heavy overlap in the locations of the peaks from the 13 whole embryo and neonate experiments.

In the response the author states

“However, I want to clarify that the manuscript does state exactly which multiple tests were performed, helping to address the reviewer’s original concern”

I may have missed it, but I am not seeing this stated explicitly. The author does show in the supplement the range of parameter values for one parameter when holding the other one to the selected value, and vice versa. In the original submission the author used a combination of parameter values such that neither parameter value is a currently selected one. I am not seeing it stated, which combination of parameter values were explored or even that combinations other than those shown in the supplement were considered.

**Have all data underlying the figures and results presented in the manuscript been provided?**

Reviewer #1: Yes

PLOS authors have the option to publish the peer review history of their article (what does this mean?). If published, this will include your full peer review and any attached files.

Reviewer #1: No

---

## [Editor Report · Acceptance letter]

13 Jul 2020

PCOMPBIOL-D-19-01723R3 

Cross-species regulatory sequence activity prediction

Dear Dr Kelley,

I am pleased to inform you that your manuscript has been formally accepted for publication in PLOS Computational Biology. Your manuscript is now with our production department and you will be notified of the publication date in due course.

With kind regards,

Laura Mallard
